# Evaluating the COVID-19 Containment Protocol in Greek Universities for the Academic Year 2021–2022

**DOI:** 10.3390/ijerph192114363

**Published:** 2022-11-02

**Authors:** Nikolaos P. Rachaniotis

**Affiliations:** Department of Industrial Management and Technology, University of Piraeus, 18534 Piraeus, Greece; nraxan@unipi.gr

**Keywords:** COVID-19 containment and surveillance protocol, tertiary education, public health

## Abstract

The COVID-19 pandemic severely disrupted European universities’ educational process. With the vaccination rollout, in-class instruction broadly resumed beginning in September 2021. In order to mitigate the risks of SARS-CoV-2 transmission, European universities apply COVID-19 containment protocols. The aim of this paper is to evaluate the COVID-19 containment protocol that Greek universities implemented in order to fully reopen in the fall of 2021 and for the entire academic year 2021–2022. A case study was conducted at the Department of Industrial Management and Technology, University of Piraeus (Athens’ port), Greece. Data were collected from November 2021 to July 2022 and a quantitative statistical analysis (descriptive and inferential) was performed. A total of 330 unique (and 43 reinfections) COVID-19 cases were confirmed, including 241 undergraduate students, 73 postgraduate, and 2 doctoral students, 10 faculty, and 4 administrative personnel. Contact tracing reported four confirmed and eight potential cases of in-classroom transmission. The person in charge of implementing the COVID-19 containment protocol in the department ordered more than 6000 rapid tests during this period. The Department of Industrial Management and Technology at the University of Piraeus used a rigorously monitored and coordinated strategy of vaccine promotion, screening/testing, contact tracing, isolation, and quarantine in order to control COVID-19 transmission. The results show, on one hand, that the protocol’s implementation is effective and leads to in-classroom transmission minimization and, on the other hand, verify the hypothesis that the department’s confirmed COVID-19 cases are less (with a mean percentage difference of 50%) than the community’s respective 18–39 age group.

## 1. Introduction

More than two and a half years after its emergence in Wuhan, China, the global spread of severe acute respiratory syndrome coronavirus 2 (SARS-CoV-2), the highly contagious causative agent of COVID-19, continues to strain healthcare systems. By 19 July 2022, 563,608,165 confirmed cases of COVID-19, including 6,371,423 deaths, were reported globally [1]. Greece over the same period recorded 4,210,771 confirmed cases of COVID-19 and 30,707 deaths [2] for a population of around 10.5 million; this performance ranks unfavorably among advanced countries and represents a major national shock with significant medical and socioeconomic impacts. Up to December 2020, due to the absence of effective therapies and vaccines, the Greek government options were limited to non-pharmaceutical interventions (NPIs) to deter viral transmission, minimize death rates, and resume normal activities. They included, inter alia, social distancing and lockdowns, travel restrictions, face-masks, teleworking, contact tracing, isolation/quarantine, and school/university closures [3].

Focusing on universities, SARS-CoV-2 posed significant challenges to their daily activities. Universities represent a unique environment with a dense population of primarily young students [4]. With the vaccination rollout, in-class instruction resumed in Greek universities in September 2021, applying a COVID-19 surveillance/containment protocol so as to mitigate the risks of SARS-CoV-2 transmission. This includes a combination of vaccination promotion and NPIs such as indoor mask-wearing, surveillance testing and screening, quarantine of suspect and confirmed cases, quarantine and frequent testing of close contacts, staggered class times, enhancements to classrooms’ air circulation, and enhanced hygiene and cleaning practices. The protocol aims to balance the optimum utilization of vaccine resources that gradually became available with the retention of some NPIs.

In order to assess the protocol’s effectiveness, a case study from the Department of Industrial Management and Technology at the University of Piraeus, a medium-sized urban university in Athens, is presented. The hypothesis that the confirmed COVID-19 cases in the department were systematically less than the community’s, due to the protocol’s rigorous implementation, is tested.

The remainder of this paper is organized as follows: Section 2 provides a review of the relevant literature. In Section 3, the proposed methodology of the study is described, whereas in Section 4 the results are presented. Finally, Section 5 discusses the study’s results and Section 6 concludes the paper.

## 2. Background from the Literature

Tertiary education students come from a variety of backgrounds, moving and intermixing at the university and at a wider community level. The features of a university, such as the students’ age distribution and extracurricular activities, influence the COVID-19 pandemic dynamics. COVID-19 containment and surveillance protocol’s success is of paramount importance for universities’ viability, since it was proven in the academic years 2019–2020 and 2020–2021 that e-learning cannot fully substitute students’ physical presence in a tertiary education system. However, although from the first days of 2020 COVID-19-related epidemiological literature thrived, research work focusing on evaluating the pandemic’s containment protocols used in universities is scarce and restricted mostly to the U.S.A. This research work can be divided in two categories: pre- and post-vaccines advent.

In the first category, efforts were made to assess the control plans used for reopening universities (e.g., Boston University in [4], Harvard/Boston/Duke/Northeastern in [5], Bristol (UK) in [6], Southern California in [7], Purdue in [8], Tulane in [9], Clemson in [10]); or to develop relevant susceptible–exposed–infected–recovered (SEIR) or agent-based (ABM) models in a university environment (e.g., Emory University in [11], University of Illinois in [12]). In the second category, research was focused on studying how COVID-19 containment protocols decreased transmission among university students (e.g., Saint Louis University in [13], Boston University in [14]) and to assess how the Omicron variant was established in universities (e.g., Harvard/Boston/Northeastern in [15], University of Washington in [16]).

Similar to the U.S.A., European universities implemented COVID-19 containment protocols including broad screening and testing. Due to the fact that they are not closed systems, and there is an ongoing risk of importation of the virus, they face considerable numbers of confirmed cases. But unlike U.S. universities, it seems that they have not shown, until now, the required interest in assessing their protocols. Therefore, it is necessary to provide data in order to support the hypothesis that a combination of SARS-CoV-2 vaccination and risk mitigation measures are effective for limiting the disease spread in a university environment. This is the gap that the present paper aims to address and, according to the best of the author’s knowledge, this is the first attempt to present results from the protocol’s application in Greece.

## 3. Methods

### 3.1. The Protocol

In order to resume in-class instruction in tertiary education, the Greek government passed a law in September 2021 [17]. According to this law, the participants in the educational process should either (a) have received at least fourteen days ago the vaccination for COVID-19, or (b) have received at least fourteen days ago the vaccination for COVID-19 with one dose of vaccine due to their infection from COVID-19, or (c) have been infected with COVID-19 more than twenty days ago and less than one hundred eighty days since the date of their diagnosis as a positive case, or (d) have been diagnosed negative either with a laboratory test for COVID-19 with the polymerase chain reaction (PCR) method carried out by taking an oropharyngeal or nasopharyngeal swab, or with a rapid test for the detection of the antigen of COVID-19. Full vaccination was defined as having received either a single dose of Janssen (Johnson & Johnson) vaccine or the second dose of Moderna or Pfizer-BioNTech or Astra-Zeneca COVID-19 vaccine. Partial vaccination was defined as receipt of one dose of Moderna or Pfizer-BioNTech or Astra-Zeneca COVID-19 vaccine. The laboratory disease control (screening test) was carried out two times a week up to forty-eight hours before every Tuesday and Friday, respectively, from 1 October 2021 until 1st May 2022 for the non-vaccinated students and personnel. After 1st May 2022, only the non-vaccinated personnel carried out a test once per week.

In order to participate in the educational process in person, students were required to bring, in printed or digital form, a certificate proving compliance with one of the above conditions, as well as a relevant identification document such as a student identification card or police identity card or passport, which was shown to the competent control bodies designated by universities at central entrances and/or auditorium entrances.

The use of a protective mask was mandatory, for all persons: (a) in all places where an educational process was carried out with physical presence, (b) in all internal spaces of the university and internal common areas of the student residences, and (c) in all external areas of the university where overcrowding was observed. Mask use was enforced for all students, staff members, and visitors. Faculty and staff members asked those who were unmasked or improperly wearing a mask to comply. Non-compliant students received sanctions, including being unable to attend classes in-person. 

The monitoring of compliance by all individuals who participated in the educational process of universities with a physical presence took place through an electronic platform (www.edupass.gov.gr, accessed on 20 July 2022). Whenever a positive laboratory test was carried out, the system was updated and the case was recorded, provided that the case was registered in the platform.

Applying this law, Greek universities were obliged to implement a COVID-19 surveillance/containment protocol, based on Greek National Public Health Organization guidelines, adapted for their academic programs. The University of Piraeus, a mid-sized public urban university in Piraeus (Athens’ port), Greece, with ten departments and approximately 17,500 undergraduate students, drafted its COVID-19 surveillance/containment protocol at the end of September 2021 and updated it at the end of April 2022. The protocol was posted on the University’s main website [18].

According to this protocol, a person in charge of implementing the COVID-19 surveillance/containment protocol in each department was assigned. Confirmed cases with COVID-19 stayed in quarantine for 10 days until 1 January 2022 and 5 days since then, based on evolving data from European Centre for Disease Prevention and Control(ECDC). Their isolation ended after this period, and if there were no symptoms with complete subsidence of fever for a 24 hour period without the use of antipyretics. In order to return to the educational process, a laboratory confirmed negative test should be carried out and after 1 January 2022, the use of a high respiratory protection mask or a double mask for at least another five days from the end of isolation was mandatory. For every confirmed case, a risk assessment for her/his possible contacts was carried out by the person in charge for the protocol’s implementation. A contact of a case of COVID-19 infection was defined as a person who had a history of contact within a period of time ranging from 48 h before the onset of symptoms of the case to 10 days (5 days since 1 January 2022) after the onset of symptoms. Depending on the level of exposure, the contacts of the case were categorized into: (a) close contacts (high-risk exposure), i.e., individuals who had face-to-face contact with a COVID-19 confirmed case at a distance less than 1.5 m for at least 15 min and without mask use by both; (b) contacts (low-risk exposure), i.e., individuals who had face-to-face contact with a COVID-19 confirmed case within 1.5 m for less than 15 min. Additional measures, such as enhanced hand/respiratory hygiene, social distancing recommendations, disinfection of classrooms/laboratories/equipment and common areas (especially the WCs), minimization of elevators’ utilization, and enhancement of all building ventilation systems, were implemented. Finally, the educational activity was carried out without taking a break, in order to reduce crowding when leaving and entering the lectures’ halls.

### 3.2. Case Study 

The Department of Industrial Management and Technology is one of the ten departments of the University of Piraeus. For the academic year 2021–2022, it enrolled 1266 undergraduate students in a four year curriculum, 197 M.Sc. students in a three semesters program, 29 Ph.D. students, 25 faculty members/adjunct faculty, and 5 administrative staff. Out of the 1266 Department’s undergraduate enrolled students, 447 were registered more than eight years ago, therefore, they are actually inactive and they do not attend courses. The same holds for the M.Sc. program, where from the 197 students, only 135 are actually active, being enrolled no more than three years ago.

According to a Department’s General Assembly Decision, at the end of September 2021, the author was assigned as the person in charge for implementing the University’s COVID-19 surveillance/containment protocol. Confirmed cases were identified by a rapid or PCR test through either *screening* the non-vaccinated students, staff, and faculty (regardless of the presence of symptoms) twice per week; or *testing*, where students, staff, and faculty were either symptomatic and tested or they were tested after an announcement from the person in charge for the COVID-19 protocol’s implementation. His information channels were the electronic platform or a direct notification by the confirmed case with an e-mail or a phone call. The confirmed cases were interviewed via phone calls, the relevant data from the confirmed cases were recorded after their informed consent and the quarantine procedure was established. Data included sex, affiliate status (undergraduate/graduate students and their enrollment year vs. faculty and administration staff), date of their positive laboratory test, symptoms’ (if any) onset date and description, contact tracing information (classes they attended the last 48 h before their symptoms’ onset and close contacts identification), and vaccination status. These data were then de-personalized. Figure 1 illustrates the COVID-19 confirmed cases’ management protocol in a flowchart.

Statistical analysis was performed using the MS Excel. The data in this study were analyzed using simple descriptive statistics and inferential statistics hypotheses’ testing, where a significance level of 5% was used.

## 4. Results

In-class instruction resumed in the Department of Industrial Management and Technology on 11 October 2021 for the graduate programs courses and 18 October 2021 for the undergraduate program courses. At that time, the department’s non-vaccinated students, undergraduate and graduate, were less than 25%, whereas faculty and staff were all fully vaccinated. From the pandemic’s start until the end of September 2021, the confirmed cases with SARS-CoV-2 infection in the department were approximately 9%.

Although it was not mandatory, students’ registration in the COVID-19 monitoring electronic platform was strongly suggested and encouraged. In total, 594 undergraduate (72.5% of the total active undergraduate students), 114 M.Sc. (84.4% of the total active M.Sc. students), 15 Ph.D. (51.7% of the total Ph.D. students), and all faculty and administrative staff were registered.

The first confirmed case after in-class instruction resumed was recorded on 4 November 2021. Since then, and until 20 July 2022, 330 unique (and 43 reinfections) COVID-19 cases were confirmed, including 241 undergraduate students, 73 postgraduate, and 2 doctoral students, 10 faculty, and 4 administrative personnel. Out of the 330 unique COVID-19 confirmed cases, 201 were males and 129 were females. Considering that females in the department account for approximately 38.7%, they had a slightly higher infection rate (44.3%) than males (43.6%).

When comparing infections by their year of studies in undergraduate students enrolled after 2018 (i.e., the ones that attend courses regularly) the percentages of those who tested positive for SARS-CoV-2 are similar, ranging from 40.9% to 45.6%. An impressive 72.7% of the first-year M.Sc. students were infected (in the second year, this figure drops to 55.9%). The percentage of confirmed cases among faculty was 40% and among administrative staff, 80%.

From the 330 unique cases with positive test results, complete data became available for 259 (the remaining 71 cases were recorded during Christmas holidays, where contact tracing was not performed). A total of 20.7% of the cases were non-vaccinated, 1.4% were partially vaccinated, 35.9% were fully vaccinated, and 42% were vaccinated with a booster dose. The total number of reinfections was 43 (21 females and 22 males, 11.5% of the confirmed cases). A total of 23.3% were non-vaccinated, 4.7% partially vaccinated, 37.1% fully vaccinated, and 34.9% vaccinated with a booster dose.

The “information delay” between the date of positive laboratory test and the person in charge’s notification had a mean of 1.3 days (s.d. = 1.3 days, min = 0, max = 8 days). The time interval between symptoms onset and the positive test result had a mean of 1.7 days (s.d. = 1.3 days, min = 0, max = 7 days). Their reported symptoms included fatigue, fever, headache and body ache, sore throat, cough, and runny nose. The time interval between symptoms onset and the person in charge’s notification had a mean of 2.9 days (s.d. = 1.6 days, min = 0, max = 10 days). 

For all 259 confirmed cases with available data, a class roster analysis was initially performed in order to identify potential close contacts and sources of transmission for 48 h before their symptoms’ onset. This practice was quickly abandoned, since it was not possible to identify every single encounter where the case was within a distance of 1.5m of each other for at least 15 min. Moreover, there was no obligatory attendance in courses’ lectures; therefore, it was impossible to verify which students attended each course. Finally, it was systematically observed that students used their masks properly only in classrooms and not in the remaining internal spaces and common areas. Therefore, it was decided that all students and faculty attending a class with a confirmed case within 48 h from symptoms’ onset were considered as close contacts. They were required to carry out a rapid or PCR test, at days 1 and 5 after the person in charge was notified until 1 May 2022 and day 5 since 1 May 2022. In total, 152 classes were mandated to carry out a test until 20 July 2022, summing up to more than 6000 tests. Table 1 summarizes the main descriptive statistics presented.

## 5. Discussion

In December 2021, COVID-19 case counts rose rapidly in Greece, with viral genomic sequencing confirming the Omicron variant as the cause. Although the University of Piraeus and the urban environment in which it is located experienced high Delta transmission at the time of Omicron introduction, Omicron rapidly became the dominant variant. Omicron’s spread through the department is documented during the Christmas holidays (23 December 2021–5 January 2022), leading to unprecedented increases in the confirmed case counts. In general, confirmed cases numbers increased after every holiday (Christmas break, one week after Greek Halloween and Ash Monday at the beginning of March and the Greek Independence Day on 25 March 2022), concentrated mostly among undergraduate/graduate students. All of these outbreaks were quickly controlled when students returned to classes. This is the first evidence that the containment protocol applied in the department was efficient and effective.

A second argument regarding the protocol’s efficacy stems from the fact that the confirmed cases in the department were systematically less than the community’s. Figure 2 illustrates the confirmed weekly COVID-19 student cases and the mandated weekly classes laboratory tests in the Department of Industrial Management and Technology, University of Piraeus, vs. the confirmed weekly COVID-19 cases in Greece for the age span 18–39, adjusted for the same population basis (4 November 2021–20 July 2022) [2]. Graphically, only in 5 out of 37 weeks were there more confirmed cases in the department than in the community. After performing an F-test of variances’ equality confirming that their variances are equal (*p*-value = 0.203), a two-sample t-test was performed comparing the means values of weekly students’ confirmed cases in the department vs. the community’s weekly confirmed cases in the age group 18–39. The yielded results (*p*-value = 0.038, mean percentage difference = 50%, 95% CI of the mean percentage difference = [0.21, 2.8]) support the argument that the hypothesis the department’s confirmed COVID-19 cases were less than the community’s respective 18–39 age group cannot be rejected at a 5% significance level.

In general, there is a lack of empirical data in a university environment about the efficacy of the individual NPIs involved, but it is beyond doubt that they reduced COVID-19 transmission. Focusing on mask usage, some first results indicate a reduction between 2.7 and 3.6 times in new infections through classroom interactions [7,13]. The extreme usefulness of mask-wearing was proven in the examined case, since during the time period it was mandatory, the rise in the Omicron variant did not increase the risk of in-class transmission and there was no concrete evidence that any confirmed cases were infected inside the classroom. More specifically, potential instances of in-class transmission were defined as two or more SARS-CoV-2 confirmed cases that shared an in-person class. Until the end of obligatory masking on 1 June 2022, only 8 out of 330 confirmed cases were deemed potential in-class transmission events. From 1 June until 16 June, when masks became mandatory again during the exam period, there were four confirmed instances of in-classroom transmission among students, whereas faculty and staff were always infected outside of the university. 

Finally, vaccine effectiveness against symptomatic Omicron infection was reduced, even in the highly vaccinated department’s population. Current vaccines are protecting against severe illness, hospitalization, and death [3], but Omicron transmission is clearly possible among both vaccinated and unvaccinated individuals, partially evading immunity acquired from prior COVID-19 infection and from a two or three dose mRNA vaccine regimen [15]. This was observed in the department, especially after the advent of Omicron variant BA.5. in June, 2022.

Summarizing, the case illustrates that an Omicron peak took place during a time when classes were not in session, and the return to class in 2022 was marked by rapidly falling disease incidences in the department, despite in-person classes, confirming evidence from the respective studies in U.S. universities [14,15,16]. Moreover, the department’s in-class instruction is not an appreciable source of COVID-19 transmission only under the setting of mandatory masking, which is in agreement with the relevant findings in the literature [14].

## 6. Conclusions

Although the major limitation of the present analysis is the inherent subjectivity of the confirmed cases remembering all relevant information during the phone interviews, the Department of Industrial Management and Technology of the University of Piraeus case is important. The short information time of the person in charge for the protocol’s implementation regarding confirmed cases, followed by rapid contact tracing and isolation, led to limited transmission in the department. No major outbreaks were observed when classes were in session and the resulting number of confirmed cases was manageable under the applied NPIs framework. It has to be stated here that students were adherent and positive to the implemented protocol, which is in agreement with previous research findings [19].

Data support the hypothesis that developing a communication channel to students, staff, and faculty; vaccination promotion; frequent and adaptive testing; and vigorous face masking prevents widespread outbreaks of COVID-19 in universities, despite the worsening epidemiological conditions. The department’s confirmed COVID-19 cases were less (with a mean percentage difference of 50%) than the community’s respective 18–39 age group. The presented evaluation can be helpful in terms of forming a standardized protocol for tertiary education institutes in case of future epidemics.

## Figures and Tables

**Figure 1 ijerph-19-14363-f001:**
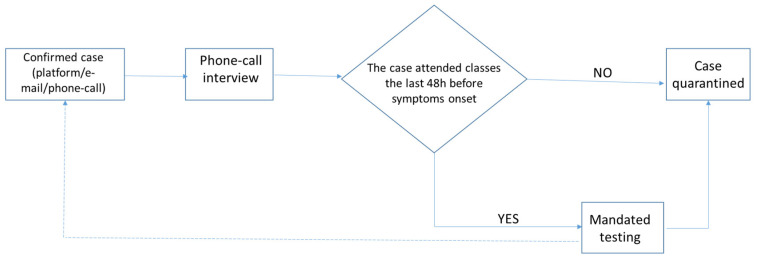
COVID-19 confirmed cases’ management protocol flowchart.

**Figure 2 ijerph-19-14363-f002:**
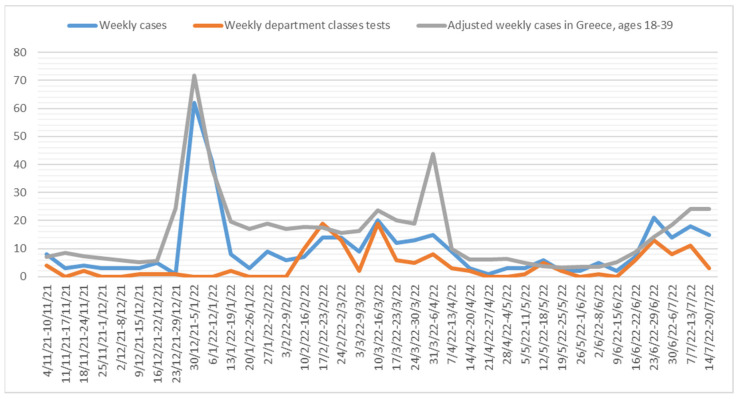
Confirmed weekly COVID-19 student cases and mandated weekly classes tests in the Department of Industrial Management and Technology, University of Piraeus, vs. confirmed weekly COVID-19 cases in Greece for the age span 18–39, adjusted for the same population basis (4 November 2021–20 July 2022).

**Table 1 ijerph-19-14363-t001:** Descriptive statistics.

Population					
		Total	Males	Females	
**Students**	Undergraduate	1266	842	424	
	M.Sc.	197	127	70	
	Ph.D.	29	21	8	
**Faculty**		25	19	6	
**Staff**		5	0	5	
**Natural History—Clinical Findings**					
Unique confirmed cases			Sex	Vaccination status (%) for confirmed cases with available data	
**Students**	Undergraduate	241	Males: 201 Females: 129	Non-vaccinated: 20.7% Partially vaccinated: 1.4% Fully vaccinated: 35.9% Booster dose: 42%	
	M.Sc.	73			
	Ph.D.	2			
**Faculty**		10			
**Staff**		4			
*Reinfections*	Total: 43		Males: 22 Females: 21	Non-vaccinated: 23.3% Partially vaccinated: 4.7% Fully vaccinated: 37.1% Booster dose: 34.9%	
Weekly Confirmed Cases and mandated classes laboratory tests					
	**Mean**	**Median**	**s.d.**	**min**	**max**
Confirmed weekly COVID-19 department’s student cases	10.1	7	11.7	1	62
Confirmed weekly COVID-19 cases in Greece, age span 18–39, adjusted in the same population basis	15.4	14.3	13.5	3.2	71.6
Weekly mandated classes laboratory tests	4	2	5.3	0	19
**Information Time Intervals (in days)**					
	**Mean**	**Median**	**s.d.**	**min**	**max**
“Information Delay”	1.3	1	1.3	0	8
Time interval between symptoms onset and a positive test result	1.7	1	1.3	0	7
Time interval between symptoms onset and the person in charge notification	2.9	3	1.6	0	10

## Data Availability

Data is available upon request.

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
