# Peer review of "Evaluating the COVID-19 Containment Protocol in Greek Universities for the Academic Year 2021–2022"

_ijerph, 2022, doi:10.3390/ijerph192114363_

Round 1

Reviewer 1 Report (Previous Reviewer 1)

1) The last sentence of the abstract is confusing. It should be formulated in a quantitative way. The 'systematically less' is not very clear.

2) The colour blue in the Figure 1 should be removed. It is hard to read the text in the figure.

3) Conclusions present a low narrative, since they should summarize the results of the manuscript.

4) The discussion is still low in comparing the case study in Greece with other cases, protocols, similar cases, etc. Try to add almost five new references for similar studies.

Author Response

Reviewer 2 Report (New Reviewer)

Thank you for an interesting paper on the COVID-19 containment protocol in a Greek university. Here are some items of inquiry that I feel the authors need to enhance:

1. Methodology - Kindly identify what is the method used in the study. In the first part of methodology section, the author/s  explained the process of the protocol. However, the type research and its method must be clearly identified at the beginning statement.

2. Bullet points - I suggest that bullet points be removed in the Section 3: Methods. These are statements explaining the process. Maybe instead of bullet points, the author can summarize the process and place it in a table. If not, just delete the bullets.

3. In the abstract it was pointed that the "aim of this paper is to evaluate the COVID-19 containment protocol that 11 Greek universities implemented in order to fully reopen in the fall of 2021 and for the entire academic year 2021-2022." It was also mentioned that the evaluation was effective. However, in the conclusion, there was no statement about the concrete result of the study. Maybe a clear statement to address the objective of the study be included in the conclusion part.

Round 2

Reviewer 1 Report (Previous Reviewer 1)

To me, the discussion is still missing on the significance of the study. It should be fully contextualized and compared with similar studies, which is not the case.

Please, produce a more critical discussion (last paragrpah). 

Author Response

1.1 To me, the discussion is still missing on the significance of the study. It should be fully contextualized and compared with similar studies, which is not the case. Please, produce a more critical discussion (last paragrpah). 

Answer 1.1. As it is mentioned in the Background literature Section, there is a handful of U.S. universities case studies and only one from a European University. The manuscript’s case study is confirming some of their findings, as stated in the last paragraph of the Discussion Section in the revised manuscript. The case’s significance is highlighted in the Conclusions Section.

This manuscript is a resubmission of an earlier submission. The following is a list of the peer review reports and author responses from that submission.

Round 1

Reviewer 1 Report

The manusript provides a case study on a COVID-19 protocol implemented in a the Greek Universities.

The author claim that they are supporting a hypothesis, but that hypothesis is not clearly presented.  I do not see a clear objective of the research.

The methodology is very weak. It seems that the idea is to develop a communication channel with the university individuals versus the out-universitie conditions, but no comparison is made on the two supporting conditions.

The establisshed conditions and analysis is very weak in presenting significant data with a solid statistical analysis. The manuscript is just a summary of conditions without a clear methodology towards solving the hypotesis.

I cannot distinguish what is the relevant data or parameters to fullfill the hypotesis. This manuscript needs a deep analyis of data with medians, standard deviations, studies on significances, etc.

Reviewer 2 Report

Dear Editor(s),

thank you for giving me the opportunity to revise this manuscript. The authors submitted a case study for assessing a COVID-19 containment protocol at the Department of Industrial Management and Technology at the University of Piraeus (Athens, Greece).

As an overall assessment, I congratulate the author on this well written article. However, it should be noted that this article lacks of novelty and it is hard to see any prospective application. Fortunately, we are now well beyond the containment phase of the pandemic. Therefore, the reader may find it uninteresting to read the article. It was to be published at least 1 year ago, I think.

Briefly, this is the major limitation that requires me to express an overall Recommendation of "reject".

There are several issues that the author should justify or solve:

1. Why doesn't the article feature other authors? A protocol should be drawn up by different professional figures.

2. The author wrote another paper on the topic [Rachaniotis NP, Dasaklis TK, Fotopoulos F, Tinios P. A Two-Phase Stochastic Dynamic Model for COVID-19 Mid-Term Policy Recommendations in Greece: A Pathway towards Mass Vaccination. Int J Environ Res Public Health. 2021 Mar 3;18(5):2497. doi: 10.3390/ijerph18052497. PMID: 33802501; PMCID: PMC7967634.] and [Rachaniotis NP, Dasaklis TK, Fotopoulos F, Chouzouris M, Sypsa V, Lyberaki A, Tinios P. Is Mandatory Vaccination in Population over 60 Adequate to Control the COVID-19 Pandemic in E.U.? Vaccines (Basel). 2022 Feb 18;10(2):329. doi: 10.3390/vaccines10020329. PMID: 35214788; PMCID: PMC8880699.].Therefore, it seems like a further fragmentation of an initial project. This is another problem that may discourage readers (as well as Editors and publisher). Notably, the second paper offers perspectives but his first study is not even mentioned.

3. The quality of the figure must necessarily be improved.

4. A flowchart of the study should be added.

5. The quality of the reference list is poor: number, scientific weight, and presentation (please refer to the editorial norms).

6. The methods are unclear (as is the main objective). The article lacks an organic structure, the discussion (but also the introduction) are often difficult to follow. For example, case analysis, testing approaches and welfare policies have been the subject of multiple studies worldwide that are not mentioned.

7. The data are described in a confused form, perhaps it would be appropriate to produce tables and graphs. It would also be appropriate to mention in the methods all the assessments made.

8. Statistics? For example, since you are making a comparison with 2018, then please present me the results in statistical manner.

9. You stated "Informed Consent Statement: Not applicable.". Nevertheless, I have a concern. You used data and made phone calls: were the subjects aware of the data collection? The sentence/methodology "These data were then de-personalized in order to present the results" is not enough.

10. You used data and made phone calls: were the subjects aware of the data collection?

Reviewer 3 Report

The proposed work aimed to present how Greek universities fully reopened in fall 2021 and assess the COVID-19 containment protocol for the academic year 2021-2022. The subject is quite interesting to its readership. Despite the notion that the work is well-described and provides valuable findings, there are a few aspects that should be taken into consideration before publishing in IJERPH. My specific comments are given below:

Within the abstract, the missing part is the conclusion, what did the author conclude by conducting this analysis? How the results are valuable to the current body of literature. At least one statement could boost the quality of the abstract overall.

Keywords are not sufficient enough.

The major point that I have noticed in the whole paper, a maximum of the acronyms is used without defining them in the first place. Please revise.

The introduction part is very weak. The arguments need to be built in the context of prior studies which is missing. Support your research questions by identifying the gaps in the literature. The contribution also seems frail. Please revise.

The methods section is strong enough and well-composed. I didn’t find any frailty within it.

The findings provided in the results require a table for clarity. The author has interpreted the results without providing them anywhere in the paper. The figure is not a professional way of describing a study’s findings. Sum up all the findings in tabular form.

In the discussion section, the results are not sufficiently endorsed by the prior literature. Discuss the work of others to support your study findings whether in favor or contrary.

There are plenty of grammatical errors and sentence structuring issues that require a professional to be fixed.  

Consider my comments positive. The manuscript is very interesting and well-presented. My suggestions will let you improve your manuscript to a significant level. Good luck with your revision.